# Prevalence and Antimicrobial Resistance of Typhoid Fever in Ghana: A Systematic Review and Meta-Analysis

**DOI:** 10.3390/diseases13040113

**Published:** 2025-04-14

**Authors:** Frederick Kungu, Aaron Awere-Duodu, Eric S. Donkor

**Affiliations:** Department of Medical Microbiology, University of Ghana Medical School, Accra P.O. Box KB 4236, Ghana; frederickungu@gmail.com (F.K.); aawereduodu@gmail.com (A.A.-D.)

**Keywords:** typhoid fever, *Salmonella* Typhi, antimicrobial resistance, prevalence, Ghana, systematic review, meta-analysis

## Abstract

**Background/Objectives**: Typhoid fever caused by *Salmonella enterica serovar* Typhi remains an important public health problem in Ghana. Understanding the epidemiology and antimicrobial resistance patterns of *S.* Typhi is crucial to guide the treatment and control of typhoid fever. This systematic review and meta-analysis aimed to estimate the prevalence of typhoid fever in Ghana and describe the antibiotic susceptibility profiles. **Methods**: Literature searches were conducted using the PubMed repository and three databases: Scopus, Web of Science, and ScienceDirect. Observational studies reporting typhoid fever prevalence among Ghanaian participants diagnosed by culture or Widal test and published between 1 January 2004 and 16 August 2024 were eligible. Quality was assessed using standardized JBI critical appraisal checklists. Random-effects meta-analysis with a 95% confidence interval was performed to estimate pooled prevalence and conduct subgroup analyses. **Results**: A total of 22 studies involving 228,107 participants were included in the systematic review. The pooled prevalence of typhoid fever was 4.14% (95% CI: 2.78–5.75). Blood culture detected more cases (3.68%) than stool culture (1.16%). Multidrug resistance was documented in 20–66% of isolates, and ciprofloxacin had the lowest prevalence of resistance (0–17%). **Conclusions**: This review highlights the substantial typhoid fever burden and evolving antimicrobial resistance in Ghana. Continuous surveillance of the disease is warranted to optimize empiric treatment and control strategies, given the resistance to first-line drugs. Enhanced prevention through water, sanitation, and vaccination programs is imperative.

## 1. Introduction

The growing burden of typhoid fever in low- and middle-income countries continues to pose a significant threat to public health [1,2], straining already overburdened healthcare systems and exacting a devastating toll on vulnerable populations who are disproportionately affected by this preventable and treatable disease [3]. The disease, caused by *Salmonella enterica serovars* Typhi and Paratyphi, causes a host of clinical symptoms, including high fever, abdominal pain, and gastrointestinal disturbances [4]. *S.* Typhi is the most common cause of the disease, while *S.* Paratyphi results in milder and less frequent infections [5]. Recent estimates have reported 9.2 million global cases of typhoid fever and 110,000 deaths, with Africa ranking third in incidence rates (111 cases per 100,000 persons), following Southeast Asia and the Eastern Mediterranean (306 and 187 cases per 100,000 persons, respectively) [6]. Similarly, surveillance across 10 sub-Saharan African countries found incidence rates as high as 383 cases per 100,000 person-years in one country (95% CI, 274–535) [7]. These regions, primarily low- and middle-income countries (LMICs), are especially vulnerable to *Salmonella* Typhi and Paratyphi transmission due to poor sanitation, inadequate hygiene, and lack of clean water [8]. Ghana faces similar challenges and this is contributing to the growing burden of typhoid fever in the country. As a result, typhoid fever accounts for an estimated 3.2% of all hospital infections [9]. A study of typhoid fever in the Volta region of Ghana from 2012 to 2016 reported a sharp rise in the incidence from 148 per 100,000 persons in the first year to 943 per 100,000 in the fourth year [10]. These alarming rates suggest that the true burden of typhoid fever in Ghana is likely to be underestimated.

Determining the prevalence of typhoid fever and identifying factors influencing its spread in Ghana are complicated by various challenges. These include inadequate diagnostic infrastructure and unreliable data sources. Although hospital databases provide some insight, they often fail to report specific serotypes, rely on the Widal test—which is prone to false positives—and exclude asymptomatic carriers [11], limiting the accuracy of prevalence estimates and complicating public health interventions.

Despite limited population-based data, available studies have offered valuable insights into the epidemiology of typhoid fever in Ghana. The growing threat of antimicrobial resistance (AMR) shown by such studies further highlights the need for integrated susceptibility data to guide treatment protocols [9]. Additionally, variations in typhoid fever diagnostic methods across studies, and geographical disparities of the disease, call for a systematic review to evaluate the impact of diagnostic methods on reported prevalence and the identification of high-risk populations, respectively. This review, therefore, addresses these gaps and provides a comprehensive understanding of typhoid fever in Ghana, focusing on variations in prevalence across different participant types and diagnostic methods as well as antibiotic susceptibility profiles.

## 2. Materials and Methods

The overall review approach was based on the condition context-population (CoCoPop) review method. Each review section was completed and reported according to the Preferred Reporting Items for Systematic Reviews and Meta-Analyses (PRISMA) guidelines [12]. The Preferred Reporting Items for Systematic Reviews and Meta-Analyses extension for Scoping Reviews (PRISMA-ScR) Checklist is included in the Appendix A.

### 2.1. Search Strategy

The literature search was performed during the period of 1–16 August 2024. The search included studies published in English and conducted within Ghana between 1 January 2004 and 16 August 2024. Original research articles describing the proportion of cases of typhoid fever and their antimicrobial susceptibility status, derived from cross-sectional, longitudinal, and cohort studies, were identified from the PubMed repository and the Scopus, Web of Science, and Science Direct databases. The search was conducted using specific keywords; terms within the same concept were connected with Boolean “OR” and combined with other components of the search terms using “AND.” The specific search terms, including keyword combinations for the three databases, were as follows:

Scopus: (“typhoid fever” OR “*Salmonella* Typhi” OR “*S.* Typhi”) AND “Ghana”

ScienceDirect: (“typhoid fever” OR “*Salmonella* Typhi” OR “*S.* Typhi”) AND “Ghana”

Web of Science: (“typhoid fever” OR “*Salmonella* Typhi” OR “*S.* Typhi”) AND “Ghana.” The search in PubMed included MeSH terms (“Typhoid Fever/blood”[Mesh] OR “Typhoid Fever/complications”[Mesh] OR “Typhoid Fever/diagnosis”[Mesh] OR “Typhoid Fever/epidemiology”[Mesh] OR “Typhoid Fever/microbiology”[Mesh] OR “Typhoid Fever/parasitology”[Mesh] OR “Typhoid Fever/prevention and control”[Mesh] OR “Typhoid Fever/transmission”[Mesh]) AND Ghana”). Reference lists of included studies and relevant reviews were also manually searched to identify any additional publications not indexed in the databases. All publications were imported into the Rayyan online platform (Rayyan—Intelligent Systematic Review, available at www.rayyan.ai, accessed on 19 August 2024) and deduplicated prior to the study selection process.

### 2.2. Study Selection Process

Two reviewers (F.K. and A.A.-D.), screened all records in the Rayyan online platform to assess eligibility based on pre-defined criteria. References were coded as “Included,” “Excluded,” or “Maybe.” Records coded as “excluded” were not further assessed. Full texts of studies labeled as “included” or “maybe” progressed to the next screening stage. A calibration exercise was performed among reviewers to ensure consistency in applying inclusion/exclusion criteria. Full texts of potentially relevant studies were exported to the Zotero reference manager and assessed independently against the eligibility criteria by two reviewers. Reasons for final exclusion of eligible articles at this stage were recorded. Any discrepancies in study selection were discussed with a third reviewer until a consensus was reached. All included studies proceeded to the data extraction stage. The inter-rater reliability among reviewers demonstrated moderate agreement, as indicated by Cohen’s kappa value of 0.44 (Appendix A).

### 2.3. Eligibility Criteria

#### 2.3.1. Inclusion Criteria

Study design: Cross-sectional, cohort, or case-control studies reporting original data.Participants: Studies conducted among the general population or specific subgroups (e.g., children, communities, etc.) in Ghana.Outcome: Studies reporting the prevalence of typhoid fever.Diagnosis: Studies utilizing bacteriological culture (blood/stool sample) or serological Widal test to detect *S.* Typhi or *S.* Paratyphi.Language: Studies published in English.Date: Studies published between 1 January 2004 and 16 August 2024.

#### 2.3.2. Exclusion Criteria

Study design: Reviews, case reports, editorials, abstracts without full textParticipants: Studies conducted outside Ghana and not involving human participantsOutcome: Studies reporting outcomes such as AMR patterns without typhoid fever prevalenceDiagnosis: Studies relying solely on clinical diagnosis or not differentiating typhoid from other fevers to ensure diagnostic accuracy and prevent overestimating prevalence due to misclassificationLanguage: Non-English publicationsDate: Studies published before 1 January 2004

### 2.4. Data Extraction

A standardized data extraction form was developed in Microsoft Excel (Table 1) and piloted on three randomly selected studies before full abstracting. The following key information was extracted from each eligible study:Bibliographic details: Author(s), publication year, titleStudy characteristics: Location, setting (urban/rural), period, designPopulation characteristics: Sample size, age groupsSampling and recruitment detailsLaboratory methods: Diagnostic test(s), case definitionOutcomes of interest: Reported typhoid fever prevalence, incidence rates

For studies also reporting antimicrobial susceptibility profiles, the following additional data points were collected:Antibiotics tested against *Salmonella* isolatesResistant/sensitive percentages for first-line drugsMulti drug-resistant (MDR)/Extensively drug-resistant (XDR) patterns

The two reviewers extracted the data independently, which was cross-checked by a third reviewer. Discrepancies were resolved through discussion until a consensus was reached. The extracted information was synthesized qualitatively due to heterogeneity in reported outcomes. This comprehensive data extraction process ensured all relevant information across studies was captured uniformly for subsequent analysis.

### 2.5. Quality Assessment

The methodological quality of the included studies was independently assessed by two reviewers using standardized JBI critical appraisal checklists for prevalence studies [32]. The checklist comprised tools that included the appropriateness of the sample frame, appropriate recruitment of study participants, adequacy of sample size, detailed description of study subjects and setting, comprehensiveness of data analysis to cover the entire sample, validity of diagnostic methods, reliability of standards to measure condition, the availability of appropriate statistical analysis, and the adequacy of response rate. Each checklist item was either scored “Yes,” “No,” “Unclear,” or “Not Applicable.” A “Yes” response to each question was scored 1, while “Unclear,” “No,” and “Not Applicable” were scored 0. All included studies were subjected to each of these tools, and the quality of the study (risk of bias) was determined as either “Good” (low-risk; score = 7–9), “fair” (moderate risk; score = 4–6), or “Poor” (high-risk; score = 0–3). Disagreements were discussed until a consensus was met, ensuring that the final quality analysis reflected a comprehensive understanding of the included studies’ methodologies, findings, and potential biases.

### 2.6. Statistical Analysis

Meta-analysis was performed using R Studio version 4.3.3. The software was used to estimate the proportion of typhoid fever due to *S.* Typhi based on culture and Widal diagnosis. Publication bias was assessed visually and statistically by the funnel plot and Egger’s regression test, respectively (Appendix A). The Freeman–Tukey double arcsine transformation was used to stabilize variances among studies, and the DerSimonian–Laird method was used to compute pooled prevalences. Heterogeneity between studies was evaluated using the I^2^ statistic, which has values of 0%, 25%, 50%, and ≥75%, representing no, low, moderate, and high heterogeneity, respectively. The proportions of typhoid fever and antimicrobial susceptibility were measured by random-effects meta-analysis with a confidence interval of 95%. Leave-one-out sensitivity analysis and meta-regression were performed to assess the robustness of our study selection and sources of heterogeneity, respectively. An alpha value of <0.05 was considered statistically significant.

## 3. Results

### 3.1. Search Results

The database search identified 266 articles. After duplicate detection and resolution, 185 non-duplicate articles were subjected to further evaluation. Of the 185, 145 articles were excluded based on title and abstract screening, resulting in 40 articles being subjected to detailed full-text review. After full-text evaluation, 18/40 articles were excluded for various reasons (typhoid fever not laboratory confirmed (3), no specific data on *S.* Typhi (8), reported on *S.* Typhimurium (2), population size not stated (5), leaving 22 eligible articles (Figure 1).

### 3.2. Characteristics of the Included Studies

The included studies had sample sizes ranging from 73 to 167,016. Most studies (13 out of 22, or 60%) were conducted in the Ashanti region. The Greater Accra and Northern regions each conducted two studies, while the Bono East and Volta regions each recorded one study. Additionally, one study was conducted across three regions (Western, Central, and Eastern). Two studies (9%) did not specify the location of the study site.

Ten of the 22 studies (45.5%) reported resistance to commonly prescribed antibiotics for typhoid fever in Ghana. Of these, 6 out of 10 (60%) documented MDR. A total of 228,107 participants were sampled across the studies, with 7231 confirmed positive for typhoid fever. Most studies (16 out of 22) used blood culture to diagnose *S.* Typhi, whereas stool culture was used in only one study. Four studies used blood and stool cultures, and one did not specify the diagnostic method.

The studies included four different types of participants. The most commonly reported participant types were hospitalized patients (10 studies) and febrile patients (8 studies). The remaining studies included participants with asymptomatic (3 studies) and suspected typhoid (1 study).

### 3.3. Pooled Prevalence of Typhoid Fever

The pooled prevalence of typhoid fever based on the included studies was 4.14% (95% CI: 2.78–5.75, *p* = 0) (Figure 2). The heterogeneity test indicated that all studies on typhoid prevalence were significantly heterogeneous (I^2^ = 99). Therefore, the random-effects model was used for the meta-analysis.

### 3.4. Prevalence of Typhoid Fever Based on the Type of Study Participants

Figure 3 presents the subgroup analysis of the prevalence of typhoid fever or *S.* Typhi based on the type of study participants. The prevalence of typhoid fever was 7.72% (95% CI: 3.69–13.02) in febrile patients, 2.47% (95% CI: 1.04–4.46) in hospitalized patients and 3.15% (95% CI: 0.05–9.82) in typhoid-suspected patients. Among asymptomatic carriers, the prevalence of *S.* Typhi was 4.31% (95% CI: 0.00–15.56).

### 3.5. Prevalence of Typhoid Fever Based on Culture Diagnosis Method

Figure 4 presents a subgroup analysis based on the culture techniques used in diagnosis. The proportion of typhoid fever or *S.* Typhi detected using blood culture was 3.68% (95% CI: 2.36–5.26), which was three times higher than the proportion of 1.16% (95% CI: 0.14–2.93) identified using stool culture. However, combining both techniques resulted in a higher prevalence of 5.33% (95% CI: 2.11–9.87). A study that did not specify the diagnostic method recorded the highest prevalence rate at 9.29% (95% CI: 7.24–11.5).

### 3.6. Quality of Included Studies

Generally, the quality of the included papers was high, with the majority (14) of studies scoring “good” quality (achieving 7–9 points) and the rest (8) rated “fair” quality with scores of 4–6 points (Appendix A).

### 3.7. Antimicrobial Susceptibility

The results of the included studies revealed varying levels of resistance to commonly used antibiotics for the treatment of typhoid fever (Table 2). Ceftriaxone resistance was observed in the 3–26% range, whereas chloramphenicol resistance was notably higher, ranging from 26–91%. Ciprofloxacin, a commonly used drug, showed relatively lower resistance rates, between 0% and 17%. Gentamicin resistance rates ranged from 43% to 46%, and that of ampicillin ranged from 24% to 95%. Amoxicillin/clavulanic acid resistance was reported to range between 10% and 65%. Tetracycline resistance ranged from 17% to 62%, whereas sulfamethoxazole/trimethoprim resistance ranged between 28% and 89%. Cefuroxime resistance was relatively low, ranging from 1% to 43%. Multidrug resistance was documented in 20–66% of the cases, highlighting the challenge of treating typhoid fever with standard antibiotics.

### 3.8. Meta-Regression

Our meta-regression showed that all the participant types had a positive effect on the observed heterogeneity; however, this effect was insignificant (*p*-value > 0.05). In terms of the diagnostic method used, blood/stool culture and unspecified diagnostic methods had a positive effect on the heterogeneity, while stool culture had a negative effect. These effects were insignificant as well. Meta-regression results for blood culture could not be generated in our model, possibly due to collinearity (Table 3).

### 3.9. Sensitivity Analysis

A leave-one-out sensitivity analysis was conducted to assess the impact of each included study on the pooled prevalence of typhoid fever in the meta-analysis. The point estimates ranged from 3.39% to 4.48%, while the confidence intervals showed an overlap. These overlapping confidence intervals indicate there is an insignificant difference in the point estimates. Despite the observed variability in point estimates, our study selection can be considered to be relatively robust (Table 4).

## 4. Discussion

This review examined findings related to *S.* Typhi in Ghana over the past two decades, encompassing several patient groups such as febrile patients, typhoid-suspected patients, and asymptomatic individuals. This broad representation highlights the prevalence of the disease among different demographics. Blood culture emerged as the primary diagnostic method in 16 (72%) of the included studies, whereas stool culture (with blood culture) was employed in only one study. The preference for blood culture can be attributed to the ease of sample collection and the higher diagnostic sensitivity for detecting *S.* Typhi in whole blood, making it the preferred method in clinical practice [33]. Although stool culture can be useful, especially for identifying chronic carriers, its limited application may reflect the challenges associated with stool sample collection and relatively lower yield [34]. 

Our meta-analysis estimated the pooled prevalence of typhoid fever in Ghana to be 4.1%. This suggests that approximately 4 in every 100 individuals in Ghana have typhoid fever, highlighting the need for effective public health interventions to control the disease. A 4-year study conducted in the Volta region of Ghana from 2012 to 2016 reported a typhoid prevalence of 3.6%, which included cases of *S.* Typhi as well as *S.* Paratyphi A and B [10]. In contrast, our review focused exclusively on *S.* Typhi and reported a higher prevalence, suggesting a steady increase in the prevalence of typhoid fever in Ghana over the years. The annual increase in typhoid cases in Ghana can be attributed to several factors. Poor water and sanitation systems remain a major driver of the spread of *S.* Typhi through contaminated food and water [35]. Rapid urbanization and overcrowding in cities intensify these issues by overwhelming existing infrastructure, thus increasing the risk of outbreaks [36]. The role of chronic carriers, who show no symptoms but continue to transmit the bacteria, also contributes to the sustained circulation of the infection within communities [37]. Addressing these factors requires improving access to clean water and sanitation, enhancing food safety and hygiene practices, implementing effective vaccination strategies, and strengthening healthcare systems for early disease diagnosis and treatment.

Our study in Ghana reports a higher typhoid prevalence (4.14%) than what is reported from Malawi (2.1–3.9%) [38] and Ethiopia (3.0%) [39]. Cameroon, however, reports a prevalence of (4.40%) [40], which is quite similar to that observed in this study. In comparison, our prevalence is lower than that in Nigeria (7–18%, with prevalence remaining below 10% since 2010) [41]. 

The subgroup analysis revealed significant variations in the prevalence of typhoid fever based on the type of study participants. Febrile patients exhibited the highest prevalence (7.72% (95% CI: 3.69–13.02). This coincides with a study conducted on febrile patients in Ethiopia, where as high as 30% of the total sample size tested positive for the typhoid fever antigen [42]. The high prevalence among febrile patients may be attributed to the fact that fever is a common symptom of typhoid fever, making these category of patients more likely to test positive. Hospitalized patients had a significantly lower prevalence of 2.47% (95% CI: 1.04–4.46), potentially reflecting that, this category of study participants may include patients admitted for reasons other than typhoid fever, thus diluting the proportion of true cases. This suggests that the febrile patient population might be a better target group for typhoid surveillance and control.

The prevalence of typhoid fever among asymptomatic patients was 4.31% (95% CI: 0.00–15.56), which, although higher than in hospitalized patients, displayed wide confidence intervals, indicating uncertainty in the estimate. This probably highlights the potential risk associated with asymptomatic carriers of typhoid fever and warrants more robust screening in populations at risk. The significant prevalence of typhoid fever associated with asymptomatic patients in the meta-analysis is concerning, particularly because one of the two studies on asymptomatic individuals focused on food vendors [20]. Given that typhoid fever spreads primarily through contaminated food and water, this finding underscores the serious public health threat posed by asymptomatic carriers, who may unknowingly contribute to the transmission of the disease. Additionally, the presence of chronic carriers in the population who do not exhibit symptoms can lead to the ongoing transmission of the disease.

The findings indicate that the prevalence of typhoid fever among typhoid-suspected patients (3.15%) was lower than that of febrile patients (7.72%). In contrast, another study in Ethiopia reported a prevalence of 6% among patients suspected of typhoid-, which is notably higher than the 2% prevalence found in febrile patients [39], indicating potential regional differences in disease presentation or diagnostic practices. This discrepancy highlights the importance of enhancing diagnostic accuracy and consistency across studies to better inform treatment decisions and public health strategies.

Our meta-analysis results based on diagnostic techniques also revealed variability in typhoid detection rates. Blood culture yielded a prevalence of 3.68% (95% CI: 2.36–5.26), three times higher than 1.16% (95% CI: 0.14–2.93) recorded using stool culture. This emphasizes the superior sensitivity of blood culture in detecting *S.* Typhi, as stool samples might only capture the pathogen in a subset of patients during specific stages of infection [43]. Interestingly, combining blood and stool cultures increased the prevalence to 5.33% (95% CI: 2.11–9.87), indicating that utilizing multiple diagnostic methods may improve detection rates [44]. The highest prevalence was observed in studies that did not specify the diagnostic method, at 9.29% (95% CI: 7.24–11.5). This suggests potential variability in the quality of diagnosis across different studies or populations and highlights the need for standardizing diagnostic protocols in research on typhoid fever to ensure more accurate and comparable prevalence estimates. Our meta-regression showed that neither the type of participants nor the diagnostic methods significantly impacted the heterogeneity observed in this study. Hence, the variability in prevalence data among the included studies could have resulted in the high heterogeneity observed.

The meta-analysis of AMR in typhoid fever highlighted trends across commonly used antibiotics for typhoid fever in Ghana. Ciprofloxacin had a lower resistance prevalence (0% to 17%) but still raises concerns in some regions, particularly Asia where high prevalence has been observed [45]. Ceftriaxone, a key first-line treatment drug, showed resistance prevalence from 3% to 26%, indicating emerging resistance. Gentamicin (43% to 46%) and ampicillin (24% to 95%) also showed high resistance prevalence, limiting their efficacy. Cefuroxime, with a resistance prevalence of 1–43%, could be an alternative treatment in areas where other treatments fail [46]. Chloramphenicol exhibited the highest resistance, ranging from 26% to 91%. This high resistance to the drug is not surprising, as chloramphenicol has been widely used for treating typhoid fever in the past. Therefore, resistance to chloramphenicol has gradually developed in many regions [47], including Ghana, due to extensive use. Several reports of resistance to chloramphenicol from studies in Ghana and other countries provide insight into its replacement by other antibiotics in treatment guidelines. A study conducted in India offers hope for the re-emergence of *Salmonella* isolates susceptibility to chloramphenicol, suggesting that chloramphenicol may once again be a viable treatment option [47]. Multidrug resistance, documented in 20–66% of typhoid fever cases in our review, poses a major challenge to treatment of the infection. For example, a rural study in Vietnam reported up to 90% of typhoid cases being associated with MDR, resulting in a high level of clinical relapse, intestinal perforation, and poor response to treatment [48]. These challenges lead to higher healthcare costs, increased complications, and reliance on more expensive alternatives, which strains healthcare systems, especially in low- and middle-income countries like Ghana [49]. While AMR is reported to affect countries across all regions and income levels, its impact is particularly severe in lower–middle-income countries [50], emphasizing the urgent need for antimicrobial stewardship, enhanced resistance surveillance, and preventive measures like vaccination and improved sanitation.

### Strengths and Limitations

Our review highlights the key strengths and weaknesses of *Salmonella* Typhi studies in Ghana, offering a broad view of the disease’s prevalence across diverse populations. The use of blood culture in 72% of the studies enhances diagnostic reliability because of its high sensitivity; however, relying primarily on blood cultures for typhoid diagnosis can impact the accuracy of prevalence estimates [51]. The sensitivity of blood cultures, particularly in the early stages of infection or after antibiotic use, is limited, leading to false negatives [52]. In resource-limited settings, where lab facilities may be inadequate, diagnosis can be delayed or missed. Cultural factors, such as delayed healthcare seeking and reliance on traditional medicine, which are common in African communities [53,54], further contribute to underreporting. Additionally, false positives from contamination can skew results. These challenges, coupled with the limited use of alternative diagnostic methods like stool cultures, may miss cases among asymptomatic chronic carriers [55], contributing to underdiagnosis and misreporting. This affects the true prevalence of typhoid. Furthermore, the concentration of studies in the Ashanti region (60%) limits the generalizability of findings, whereas variability in diagnostic protocols complicates data comparison. The findings from our systematic review are clinically relevant and can help guide medical decision-making by identifying high-risk groups such as febrile patients and food vendors and informing screening and treatment efforts. They also underscore the gaps in the detection of asymptomatic carriers and highlight the need for better screening and treatment strategies.

The high prevalence of typhoid fever across different patient groups emphasizes the urgent need for a comprehensive approach that addresses Water, Sanitation, and Hygiene (WASH) challenges and vaccination. This aligns with the Water Sector Development Plan (2014), which sets the Ghanaian government’s goal of achieving universal access to safe drinking water by 2025 and eliminating open defecation by 2030 [56]. Also, as part of the Typhoid Conjugate Vaccine Introduction in Africa (THECA) program, a cluster-randomized trial has been initiated in Ghana in 2021 to further support these public health efforts [3]. Despite these efforts, multiple challenges remain, including findings from Acosta-Alonzo et al. (2020), which reported that people are less likely to get vaccinated if they perceive the cost as too high or the process as difficult, making it difficult to achieve herd immunity in the population [57]. They also noted that while treatments for typhoid are available, proximity to the hospitals significantly influences how often people seek medical care [57]. Addressing these challenges requires key measures, including investing in sustainable water and sanitation infrastructure, conducting hygiene education campaigns, and integrating typhoid conjugate vaccines into routine immunization campaigns, focusing on high-risk populations. Additionally, standardizing diagnostic protocols could improve prevalence estimates and public health responses. This review opens new avenues for research, particularly into asymptomatic carriers in key populations such as food vendors, alternative diagnostic methods, and socioeconomic factors driving typhoid’s spread in different regions, and paves the way for more tailored interventions in Ghana.

## 5. Conclusions

This study revealed a significant prevalence of *S.* Typhi in Ghana, with a pooled estimate of 4.1% across various patient populations, including febrile and asymptomatic individuals. The high prevalence, particularly among asymptomatic carriers, intensifies the ongoing transmission of the disease. Our findings highlight the urgent need for improved public health interventions and more accurate diagnostic methods, as the reliance on blood culture as the primary diagnostic tool may overlook cases, indicating the need for alternative diagnostic approaches.

### Recommendation

In future research, we recommend focusing on asymptomatic carriers, especially in high-risk groups like food vendors, to capture hidden infection. Expanding diagnostic methods, such as stool cultures, could improve the identification of chronic carriers. Additionally, broadening the research scope beyond the Ashanti region and conducting longitudinal studies to monitor trends in AMR will provide a more comprehensive view of the disease across Ghana. Investigating environmental and socioeconomic factors and evaluating public health measures like vaccination could offer valuable insights into effective disease control.

## Figures and Tables

**Figure 1 diseases-13-00113-f001:**
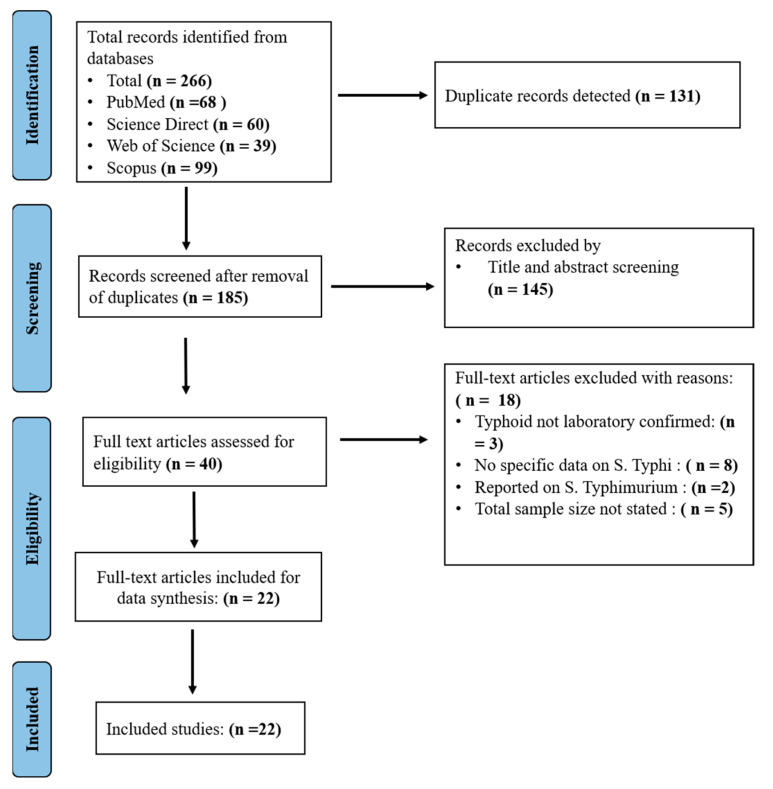
PRISMA diagram of included studies.

**Figure 2 diseases-13-00113-f002:**
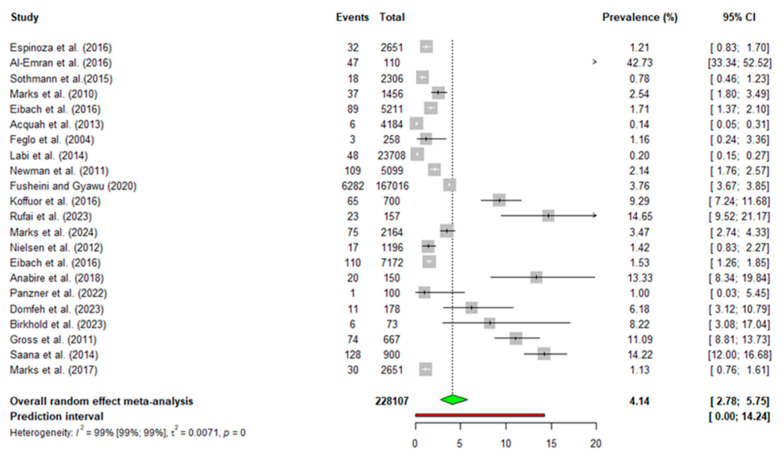
Pooled prevalence of typhoid fever from 2004 to 2024 in Ghana [9,10,11,13,14,15,16,17,18,19,20,21,22,23,24,25,26,27,28,29,30,31].

**Figure 3 diseases-13-00113-f003:**
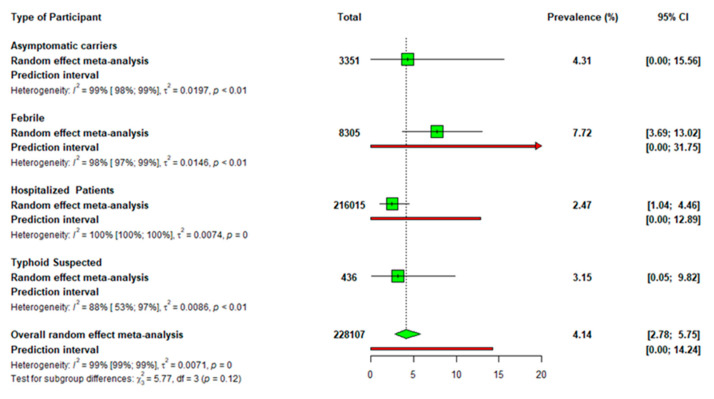
Subgroup analysis of the prevalence of typhoid fever based on the type of the study participants.

**Figure 4 diseases-13-00113-f004:**
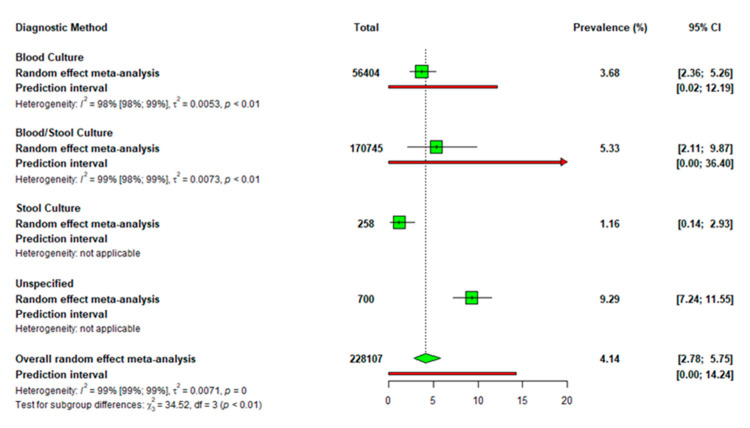
Subgroup analysis of typhoid fever prevalence based on the type of culture diagnosis.

**Table 1 diseases-13-00113-t001:** Study characteristics.

Authors	Study Design	Region	Type of Participants	Sample/Population Size	No. of Positive *S.* Typhi Isolates	Diagnostic Method
Acquah et al. (2013) [13]	Retrospective	Northern	Hospitalized Patients	4184	6	Blood Culture
Al-Emran et al. (2016) [14]	Cross-sectional	Ashanti	Febrile	110	47	Blood Culture
Anabire et al. (2018) [15]	Cross-sectional	Northern	Febrile	150	20	Blood Culture, Widal
Birkhold et al. (2023) [16]	Prospective cohort	Ashanti	Hospitalized Patients	73	6	Blood Culture
Domfeh et al. (2023) [11]	Cross-sectional	Ashanti	Typhoid Suspected	178	11	Blood Culture, Stool Culture, Widal
Eibach et al. (2016) [17]	Cohort	Ashanti	Hospitalized Patients	5211	89	Blood Culture
Eibach et al. (2016) [18]	Prospective cohort	Ashanti	Hospitalized Patients	7172	110	Blood Culture
Espinoza et al. (2016) [19]	Cohort	Ashanti	Asymptomatic individuals	2651	32	Blood Culture
Feglo et al. (2004) [20]	Cross-sectional	Ashanti	Asymptomatic individuals	258	3	Stool Culture, Widal
Fusheini and Gyawu (2020) [10]	Longitudinal	Volta	Hospitalized Patients	167,016	6282	Blood Culture, Stool Culture
Gross et al. (2011) [21]	Prospective cohort	Western, Central, Eastern	Febrile	667	74	Blood Culture
Koffuor et al. (2016) [22]	Cross-sectional	Ashanti	Asymptomatic individuals	700	65	Not indicated
Labi et al. (2014) [23]	Retrospective	Greater Accra	Hospitalized Patients	23708	48	Blood Culture
Marks et al. (2010) [24]	Cohort	Ashanti	Hospitalized Patients	1456	37	Blood Culture
Marks et al. (2017) [25]	Cohort		Febrile	2651	30	Blood Culture, Stool Culture
Marks et al. (2024) [26]	Cohort	Ashanti	Febrile	2164	75	Blood Culture
Newman et al. (2011) [27]	Prospective cohort		Hospitalized Patients	5099	109	Blood Culture
Nielsen et al. (2012) [28]	Prospective Cohort	Ashanti	Hospitalized Patients	1196	17	Blood Culture
Panzner et al. (2022) [29]	Cross-sectional	Bono East	Febrile	100	1	Blood Culture
Rufai et al. (2023) [30]	Cross-sectional	Greater Accra	Febrile	157	23	Blood Culture, Widal
Saana et al. (2014) [9]	Cross-sectional	Ashanti	Hospitalized Patients	900	128	Blood Culture, Stool Culture
Sothmann et al. (2015) [31]	Case-control	Ashanti	Febrile	2306	18	Blood Culture

**Table 2 diseases-13-00113-t002:** Percentage resistance of *S.* Typhi isolates to commonly used antibiotics.

Study	CTR	CHL	CIP	GM	AMP	AMC	TET	SXT	CXM	MDR
Eibach et al., 2016 [18]		76			70			76		66
Gross et al., 2011 [21]		91			95			89	1	
Labi et al., 2014 [23]			0						15	
Marks et al., 2010 [24]		73		46	70	65	62	70		
Marks et al., 2017 [25]		77			67	10		80		63
Marks et al., 2024 [26]	3	27	17		24	16	17	28		20
Newman et al., 2011 [27]										62
Nielsen et al., 2012 [28]		71			65	24	53	71		65
Rufai et al., 2023 [30]	26	26		43	35		39		43	
Saana et al., 2014 [9]	15	26	17		52			33		20

Abbreviations: CTR—Ceftriaxone, CHL—Chloramphenicol, CIP—Ciprofloxacin, GM—Gentamicin, AMP—Ampicillin, AMC—Amoxicillin/clavulanic acid, TET—Tetracycline, SXT—Sulfamethoxazole/trimethoprim, CXM—Cefuroxime, MDR—Multidrug Resistance. Blank space indicate resistance data not available.

**Table 3 diseases-13-00113-t003:** Meta-regression of factors affecting heterogeneity in the study.

Covariates	Estimate	Standard Error	z-Value	*p*-Value	95% Confidence Interval	
**Type of Participants**						
Asymptomatic Carriers (Intercept)	0.1109	0.0797	1.3910	0.1642	−0.0454	0.2672
Febrile	0.1615	0.0855	1.8883	0.0590	−0.0061	0.3292
Hospitalized patients	0.0414	0.0844	0.4902	0.6240	−0.1241	0.2069
Typhoid suspected	0.1095	0.1290	0.8490	0.3959	−0.1433	0.3623
**Diagnostic Method**						
Blood/Stool Culture	0.0358	0.0511	0.7002	0.4838	−0.0644	0.1359
Stool Culture	−0.1042	0.1323	−0.7876	0.4310	−0.3636	0.1551
Unspecified	0.1997	0.1139	1.7529	0.0796	−0.0236	0.4230

**Table 4 diseases-13-00113-t004:** Sensitivity analysis of included studies.

Excluded Study	Prevalence (%)	Lower CI	Upper CI
Acquah et al. (2013) [13]	4.48	3.05	6.16
Al-Emran et al. (2016) [14]	3.39	2.16	4.86
Anabire et al. (2018) [15]	3.87	2.53	5.46
Birkhold et al. (2023) [16]	4.04	2.67	5.66
Domfeh et al. (2023) [11]	4.06	2.69	5.70
Eibach et al. (2016) [17]	4.32	2.85	6.07
Eibach et al. (2016) [18]	4.34	2.85	6.11
Espinoza et al. (2016) [19]	4.35	2.90	6.07
Feglo et al. (2004) [20]	4.32	2.89	6.00
Fusheini and Gyawu (2020) [10]	4.16	2.77	5.80
Gross et al. (2011) [21]	3.87	2.53	5.47
Koffuor et al. (2016) [22]	3.93	2.58	5.55
Labi et al. (2014) [23]	4.34	3.14	5.73
Marks et al. (2010) [24]	4.24	2.81	5.93
Marks et al. (2017) [25]	4.36	2.91	6.07
Marks et al. (2024) [26]	4.19	2.76	5.88
Newman et al. (2011) [27]	4.29	2.82	6.04
Nielsen et al. (2012) [28]	4.32	2.89	6.02
Panzner et al. (2022) [29]	4.29	2.88	5.96
Rufai et al. (2023) [30]	3.83	2.50	5.42
Saana et al. (2014) [9]	3.77	2.46	5.32
Sothmann et al. (2015) [31]	4.39	2.94	6.10

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
