# Peer review of "Prevalence and Antimicrobial Resistance of Typhoid Fever in Ghana: A Systematic Review and Meta-Analysis"

_diseases, 2025, doi:10.3390/diseases13040113_

Round 1
Reviewer 1 Report
Comments and Suggestions for Authors
Thank you for the opportunity to review this interesting manuscript - a systematic review and meta-analysis on the prevalence of typhoid fever and antibiotic resistance in Ghana over the last two decades. Please kindly find point-by-point recommendation and/or suggestions below:
In the search methodology,
- Ensure transparency by including the full search strings. This would allow reproducibility of the search in future studies.
- Consider adding a brief explanation of the rationale behind each inclusion/exclusion criterion, particularly regarding the diagnostic methods, to address any potential biases that may have resulted from these choices.
In the quality assessment,
- Provide a summary table detailing specific strengths and weaknesses for each study based on the JBI criteria. This could include factors like sample representativeness, response rates, and statistical rigor.
- Clarify how studies with “fair” and “poor” quality scores influenced the pooled analysis. For example, a sensitivity analysis excluding lower-quality studies could help confirm whether the findings are robust despite variations in study quality.
In the data Extraction section,
- Add a section detailing the data extraction process, including quality control steps, such as independent cross-checking of extracted data and any specific software used for managing extracted information.
- Consider reporting inter-rater reliability (e.g., Cohen's kappa) for data extraction and study selection phases, to ensure consistency across reviewers.
In statistical Analysis,
- Address the substantial heterogeneity observed in the pooled prevalence estimates by conducting additional analyses (e.g., meta-regression), if possible, to identify possible factors driving variability across studies.
- If possible, incorporate a sensitivity analysis by recalculating prevalence estimates without outliers or lower-quality studies to test the robustness of the results.
- Also, enhance the transparency of results by including a funnel plot to visually assess publication bias, complemented by statistical tests (e.g., Egger’s or Begg’s tests).
In Discussion section,
- Compare Ghana’s typhoid fever prevalence and resistance rates to those in other Sub-Saharan African countries or LMICs. How the findings can provide a broader perspective on where Ghana stands regionally, guiding policymakers on whether the observed rates are typical or require urgent action.
- What is the implications of relying primarily on blood cultures for typhoid diagnosis. Consider adding a section on how diagnostic challenges (e.g., availability of reliable tests, cultural variations) might impact the accuracy of reported prevalence rates.
- What is the potential public health impact of the high multidrug resistance rates observed in the meta-analysis? Consider calling for increased AMR surveillance and targeted interventions in high-prevalence regions, especially in light of Ghana’s current healthcare infrastructure.
- Expand on the study’s implications for policy, particularly in areas related to improved sanitation, clean water access, and vaccination. Highlight any gaps in Ghana’s current public health strategies that could be addressed to control the spread of typhoid.
- What is more specific areas for further research investigation? Consider focusing on asymptomatic carriers in key populations (e.g., food vendors), exploring socioeconomic and environmental factors influencing typhoid prevalence in different regions of Ghana, and longitudinal studies to monitor trends in AMR and assess the impact of any new public health interventions over time.
I look forward to a much improved version in the next update.
Reviewer 2 Report
Comments and Suggestions for Authors
Detailed Comments:
1. Lines 2-3: Please revise the title to clearly indicate that the article assesses prevalence, similar to: “Prevalence of Typhoid Fever in Ghana: A Systematic Review and Meta-Analysis.”
2. Line 13: “PubMed” is not a database; please use the correct terminology.
3. Line 27: Add keywords such as “systematic review” and “meta-analysis”, etc
4. Line 76: Specify the type of “original research” conducted, as required by epidemiological standards.
5. Line 78: Again, “PubMed” is not a database. Please correct this.
6. Lines 81-82: Provide the exact search algorithm used for each database.
7. Line 110: The research question was focused on calculating pooled prevalence. However, it’s unclear how data on incidence were used to derive prevalence estimates.
8. Lines 111-112: The meta-analysis combines data from studies using different diagnostic techniques (bacteriological culture and the Widal test) on both blood and stool samples. This approach is inappropriate for deriving a reliable pooled prevalence estimate because the differences in diagnostic accuracy between these methods can introduce significant bias. Blood and stool cultures have varying sensitivities depending on the stage of infection, with blood cultures being more effective early on, while stool cultures are more sensitive in the later stages when bacteria are shed. Additionally, the Widal test is a serological method with variable sensitivity and specificity, and it does not directly detect the pathogen. Pooling data from such heterogeneous sources leads to significant heterogeneity, making the findings difficult to interpret accurately.
9. Lines 162-167: The statistical analysis section is extremely weak. Please provide a detailed description of the analytical process, including the interpretation of I² statistics. Explain why sensitivity analyses and meta-regression were not performed. Additionally, the manuscript redundantly mentions the version of RStudio used in both lines 162 and 166-167—this seems to be an error.
10. Materials and Methods: The PRISMA checklist and PROSPERO registration are entirely missing. These are essential components for systematic reviews and must be included.
11. Table 1: The author mentions that several studies in the table have “not applicable” study designs. It is unclear why these studies were included. Were interventional studies also considered along with observational ones? If so, please justify this decision. Furthermore, it’s advisable to categorize the studies by year or alphabetically rather than listing them in a seemingly random order.
12. Lines 204-205: Provide an interpretation for each I² value and correct the statement regarding “considerable heterogeneity.” Also, report the confidence interval (CI) for I². Please apply these changes consistently across all analyses.
13. Lines 221-223: Re-evaluate the quality assessment, as it is unlikely that all observational studies could achieve a high-quality rating.
Round 2
Reviewer 1 Report
Comments and Suggestions for Authors
Thank you for addressing all my previous comments, satisfactorily. All the best for your publication.
Reviewer 2 Report
Comments and Suggestions for Authors
All comments have been adequately addressed.